

# Multivariate stochastic bias corrections with optimal transport

Yoann Robin[1], Mathieu Vrac[1], Philippe Naveau[1], and Pascal Yiou[1]

[1]Laboratoire des Sciences du Climat et de l'Environnement, UMR 8212 CEA-CNRS-UVSQ, IPSL & U Paris-Saclay, Gif-sur-Yvette, France

**Correspondence:** Yoann Robin (yoann.robin@lsce.ipsl.fr)

**Abstract.** Bias correction methods are used to calibrate climate model outputs with respect to observational records. The goal is to ensure that statistical features (such as means and variances) of climate simulations are coherent with observations. In this article, a multivariate stochastic bias correction method is developed based on optimal transport. Bias correction methods are usually defined as transfer functions between random variables. We show that such transfer functions induce a joint probability distribution between the biased random variable and its correction. The optimal transport theory allows us constructing a joint distribution that minimizes an energy spent in bias correction. This extends the classical univariate quantile mapping techniques in the multivariate case. We also propose a definition of non-stationary bias correction as a transfer of the model to the observational world, and we extend our method in this context. Those methodologies are first tested on an idealized chaotic system with three variables. In those controlled experiments, the correlations between variables appear almost perfectly corrected by our method, as opposed to a univariate correction. Our methodology is also tested on daily precipitation and temperatures over 12 locations in southern France. The correction of the inter-variable and inter-site structures of temperatures and precipitation appears in agreement with the multi-dimensional evolution of the model, hence satisfying our suggested definition of non-stationarity.

*Copyright statement.* TEXT

## 1 Introduction

Global Climate Models (GCM) and Regional Climate Models (RCM) are used to study the climate system. However, their outputs often appear biased compared to observational references (e.g., Randall et al., 2007). For example, the temperature means can be shifted. Thus removing this bias is often necessary to drive impact studies such as based on crop or hydrological models (Chen et al., 2013). The main goal of bias correction (BC) is to match the statistical features of climate models outputs with observations (see, e.g. Ehret et al., 2012; Gudmundsson et al., 2012). The most used method is the quantile mapping (Panofsky and Brier, 1958; Wood et al., 2004; Déqué, 2007), which adjusts the quantiles of the variables of interest in the stationary case (Shrestha et al., 2014). The importance of the stationarity hypothesis has been discussed by a few studies (Christensen et al., 2008; Maraun, 2012; Nahar et al., 2017). Some extensions, like CDF-$t$ (Cumulative Distribution Function transfer, Michelan-



geli et al., 2009), can take into account some of the non-stationarity in GCM or RCM.

Most of those methods are univariate, and do not take into account the spatial and inter-variable correlations, which may alter the quality of the corrections (e.g., Wilcke et al., 2013; Maraun, 2016). Maraun et al. (2017) have pointed out that correcting
model output could induce biases of physical processes and that such procedures require an understanding of the nature of the biases. In particular it is crucial to investigate the way key climate variables co-vary.

This shortcoming has led to the recent development of multivariate techniques. As mentioned by Vrac (2018), two kinds of methods are currently available. The first type corrects separately each marginal, and apply afterwards a correction of the
dependence structure (e.g., Vrac and Friederichs, 2015; Vrac, 2018; Nahar et al., 2018; Cannon, 2018). The second kind performs recursive corrections: each variable is corrected conditionally on the previously already corrected variables (Bárdossy and Pegram, 2012; Dekens et al., 2017). These last methods have two main limitations. First, the correction depends on the ordering of the marginals. Second, each marginal is adjusted conditionally on previously corrected marginals, which reduces the number of data at each step. Furthermore, the variability of observations is generally greater than that of the climate models.
To increase the variability, von Storch (1999), Wong et al. (2014) and Mao et al. (2015) suggested to introduce a stochastic component in the bias correction procedure. In this paper, we develop a multivariate and stochastic bias correction method, different from the two categories presented, based on elements from the optimal transport theory.

Optimal transport theory is a natural way to measure the dissimilarity between multivariate probability distributions (Villani,
2008; Muskulus and Verduyn-Lunel, 2011; Robin et al., 2017), especially in a multivariate case. For example, this has been already successfully applied in image processing to transfer colors between images (Rubner et al., 2000; Ferradans et al., 2013). Here, our goal is to apply optimal transport techniques to perform bias correction in estimating a particular *joint law* (called a *transport plan*) that links the probability distributions of a biased random variable and its correction. This joint law minimizes a cost function, representing the *energy* needed to transform a multivariate probability distribution to another. In this optimal
transport context, any realization of the biased random variable induces a conditional law of the transport plan, associating the realization and its correction. As the corrections are randomly drawn from these conditional laws, the suggested method is stochastic by construction.

Moreover, Maraun et al. (2017) also stressed that BC methods do not correct the physical processes of the model, and errors
can propagate into the corrections. However, one key-aspect of the present work is to highlight that, in a climate change context (or more generally, in a framework where corrections are performed in conditions different from the calibration dataset) a proper BC method should provide changes - from calibration to projection periods - in agreement with the modelled data to be corrected. Knowing the quality of the raw modelled data (and of the underlying processes) is therefore an important a priori step. Nevertheless, this is out of the BC method development per se.






This paper is organized as follows. In Section 2, the developed theoretical framework to perform bias correction is presented. In particular, the classical definition of bias correction as transfer function is generalized with optimal transport theory. Two methods are presented : Optimal Transport Correction (OTC, stationary case) and dynamical Optimal Transport Correction (dOTC, non-stationary case). In Section 3, the proposed methodology is tested on an idealized non-stationary case based on

chaotic attractors. In Section 4, a multivariate bias correction is performed on a regional climate model (RCM) simulation of temperatures and precipitation, in a cross-validation case. Section 5 provides conclusions and perspectives.

## 2  Theoretical framework

### 2.1  Bias correction as a joint distribution

The general goal of this paper is the correction of a random variable, noted $\mathbf{X}$ (e.g., a biased climate model output) with respect

to a reference random variable, noted $\mathbf{Y}$. The random variables $\mathbf{X}$ and $\mathbf{Y}$ live in dimension $d$. If $d = 1$, we note them $X$ and $Y$. The probability law of $\mathbf{X}$ (resp. $\mathbf{Y}$) is noted $\mathbb{P}_{\mathbf{X}}$ (resp. $\mathbb{P}_{\mathbf{Y}}$).

Following Piani et al. (2010), a bias correction method of $\mathbf{X}$ with respect to $\mathbf{Y}$ is a *transfer function* $\mathcal{T} : \mathbb{R}^d \to \mathbb{R}^d$ such that the random variable $\mathcal{T}(\mathbf{X})$ follows the same law as $\mathbf{Y}$, i.e. $\mathbb{P}_{\mathcal{T}(\mathbf{X})} = \mathbb{P}_{\mathbf{Y}}$. This definition covers most of the practical cases,

but we can construct random variables where no transfer function exists, e.g. if $\mathbf{X}$ is constant and $\mathbf{Y}$ is not. Thus, beyond a multivariate transfer function, it is necessary to extend the definition of bias correction.

We assume momentarily the existence of a transfer function $\mathcal{T}$. By construction, the random variables $\mathbf{X}$ and $\mathcal{T}(\mathbf{X})$ are dependent, and their associated joint law can be summarized by the function $\kappa : \mathbb{R}^d \to \mathbb{R}^d \times \mathbb{R}^d$

$\kappa(\mathbf{x}) := (\mathbf{x}, \mathcal{T}(\mathbf{x})) \in \mathbb{R}^d \times \mathbb{R}^d.$

The map $\kappa$ connects the random variable $\mathbf{X}$ with its correction $\mathcal{T}(\mathbf{X})$ on the space $\mathbb{R}^d \times \mathbb{R}^d$. Furthermore, the map $\kappa$ induces a probability law on $\mathbb{R}^d \times \mathbb{R}^d$, noted $\mathbb{P}_{\mathcal{T}}$, and given for all measurable sets $A \subset \mathbb{R}^d \times \mathbb{R}^d$ by

$\mathbb{P}_{\mathcal{T}}(A) := \mathbb{P}_{\mathbf{X}}(\kappa^{-1}(A)) = \mathbb{P}_{\mathbf{X}}(\{\mathbf{x} \in \mathbb{R}^d \text{ such that } \kappa(\mathbf{x}) \in A\}) = \mathbb{P}((\mathbf{x}, \mathcal{T}(\mathbf{x})) \in A).$

The critical property here concerns the margins of $\mathbb{P}_{\mathcal{T}}$: *the first (resp. second) margin of $\mathbb{P}_{\mathcal{T}}$ is $\mathbb{P}_{\mathbf{X}}$ (resp. $\mathbb{P}_{\mathbf{Y}}$)*. To understand

why it is critical, we note $\Gamma(\mathbb{P}_{\mathbf{X}}, \mathbb{P}_{\mathbf{Y}})$ the set of probability measures on $\mathbb{R}^d \times \mathbb{R}^d$ such that $\mathbb{P}_{\mathbf{X}}$ is the first margin and $\mathbb{P}_{\mathbf{Y}}$ the second one. By definition, $\mathbb{P}_{\mathcal{T}} \in \Gamma(\mathbb{P}_{\mathbf{X}}, \mathbb{P}_{\mathbf{Y}})$. Thus, any bias correction method defined by a transfer function induces an element of $\Gamma(\mathbb{P}_{\mathbf{X}}, \mathbb{P}_{\mathbf{Y}})$.



We argue that *any probability distribution in* $\Gamma(\mathbb{P}_\mathbf{X}, \mathbb{P}_\mathbf{Y})$ *induces a bias correction method.* For $\gamma \in \Gamma(\mathbb{P}_\mathbf{X}, \mathbb{P}_\mathbf{Y})$, $\gamma(\mathbf{x}, \mathbf{y})$ can be interpreted as *the probability that* $\mathbf{y}$ *is the correction of* $\mathbf{x}$. Formally, the Jirina theorem (see e.g., Strook, 1995, chap. 5) states that there exists a collection of probability laws $\gamma_\mathbf{x}$, $\mathbf{x} \in \mathbb{R}^d$, such that $\gamma_\mathbf{x}$ are the conditional laws of $\mathbf{Y}$ *given* $\mathbf{X}$. In other words, for $B \subset \mathbb{R}^d$, $\gamma_\mathbf{x}(B)$ is the probability that the correction $\mathbf{y} \in B$, given $\mathbf{X} = \mathbf{x}$. The correction of $\mathbf{x}$ is then sampled from

the law $\gamma_\mathbf{x}$. Thus, any $\gamma \in \Gamma(\mathbb{P}_\mathbf{X}, \mathbb{P}_\mathbf{Y})$ defines a bias correction method, through the conditional laws $\gamma_\mathbf{x}$. Here, the stochastic part of this approach is highlighted. All corrections are sampled from the laws $\gamma_\mathbf{x}$, and the corrected values follow the law $\mathbb{P}_\mathbf{Y}$ (by definition of a conditional law).

We note that the problem where $\mathbf{X}$ is constant is easily solved with this approach. The set $\Gamma(\mathbb{P}_\mathbf{X}, \mathbb{P}_\mathbf{Y})$ is reduced to one element:

the independent law $\delta_\mathbf{x} \times \mathbb{P}_\mathbf{Y}$, where $\delta_\mathbf{x}$ is the Dirac mass in $\mathbf{x}$. Thus, $\gamma_\mathbf{x} = \mathbb{P}_\mathbf{Y}$, and the correction of $\mathbf{X}$ is given by sampling each correction with the law $\mathbb{P}_\mathbf{Y}$.

We have defined a bias correction method as an element of $\Gamma(\mathbb{P}_\mathbf{X}, \mathbb{P}_\mathbf{Y})$. However, this set can be very large. The goal of the next section is to present a *criterion* to select an element of $\Gamma(\mathbb{P}_\mathbf{X}, \mathbb{P}_\mathbf{Y})$.

## 2.2  Selection of a joint law with optimal transport theory

To select a probability law $\gamma \in \Gamma(\mathbb{P}_\mathbf{X}, \mathbb{P}_\mathbf{Y})$, we propose to use a cost function on this set. The minimum of this cost function corresponds to an optimal bias correction method. We propose to minimize the energy needed to transform a realization $\mathbf{x}$ of $\mathbf{X}$, to its correction $\mathbf{y}$, i.e. minimize $\|\mathbf{x} - \mathbf{y}\|^2$, weighted by $\gamma(\mathbf{x}, \mathbf{y})$. Thus, the cost function $C$ is given by :

$$C : \begin{cases} \Gamma(\mathbb{P}_\mathbf{X}, \mathbb{P}_\mathbf{Y}) & \to \mathbb{R}_+, \\ \gamma & \mapsto \int_{\mathbb{R}^d \times \mathbb{R}^d} \|\mathbf{x} - \mathbf{y}\|^2 . \mathrm{d}\gamma(\mathbf{x}, \mathbf{y}). \end{cases} \tag{1}$$

This cost function minimizes the square of the distance between $\mathbf{x}$ and its correction $\mathbf{y}$. Our bias correction method is associated with the law $\gamma$ that minimizes $C$. This cost function stems from optimal transport theory (Villani, 2008). To understand this choice, we propose to examine the univariate case. We note $F$ (resp. $G$) the Cumulative Distribution Function (CDF) of $X$ (resp. $Y$), defined by $F(x) := \mathbb{P}_X(X \le x)$ (resp. $G(y) := \mathbb{P}_Y(Y \le y)$). We assume that $F$ and $G$ are continuous. The transfer function of the quantile mapping method is given by $\mathcal{T} = G^{-1} \circ F$, where $G^{-1}$ corresponds to the quantile function of $Y$, i.e.

the inverse of $G$. To understand this definition, we can write that

$$
\begin{aligned}
\mathbb{P}_{\mathcal{T}(X)}(\mathcal{T}(X) \le y) &= \mathbb{P}_{\mathcal{T}(X)}(F(X) \le G(y)), && \text{by definition of } \mathcal{T}, \\
&= G(y), && \text{because } F(X) \text{ follows the uniform law,} \\
&= \mathbb{P}_Y(Y \le y), && \text{by definition of } G.
\end{aligned}
$$





Hence, the random variable $\mathcal{T}(X)$ follows the same law as $Y$. In the univariate case, it can be shown that the joint law defined by $\mathcal{T} = G^{-1} \circ F$ minimizes the cost function $C$ of Eq. (1). Proofs of this statement can be found in Farchi et al. (2016, Appendix A) and Santambrogio (2015, Chap. 2).

To summarize the univariate case, we have seen that the classical quantile mapping can be viewed as the solution of a mini-
mization problem based on the cost function of Eq. (1). This solution can be derived from optimal transport theory. Our next step is to explain how this minimization strategy can be extended in the multivariate case.

### 2.3   Multivariate bias correction with optimal transport selection : the stationary case

Given $(\mathbf{X}_1, \ldots, \mathbf{X}_n)$ and $(\mathbf{Y}_1, \ldots, \mathbf{Y}_n)$ two independent and identically distributed (i.i.d.) samples of the random variables $\mathbf{X}$ and $\mathbf{Y}$. A first step is to estimate the empirical distributions, $\hat{\mathbb{P}}_{\mathbf{X}}$ and $\hat{\mathbb{P}}_{\mathbf{Y}}$. We note $\mathbf{c}_i$ a collection of regularly spaced cells that
partition $\mathbb{R}^d$, and cover $(\mathbf{X}_1, \ldots, \mathbf{X}_n)$ and $(\mathbf{Y}_1, \ldots, \mathbf{Y}_n)$. The center of each cell is also noted $\mathbf{c}_i$. With this notation, $\hat{\mathbb{P}}_{\mathbf{X}}$ and $\hat{\mathbb{P}}_{\mathbf{Y}}$ can be written as a sum of $I$ and $J$ Dirac masses:

$$\hat{\mathbb{P}}_{\mathbf{X}}(A) = \sum_{i=1}^{I} p_{\mathbf{X},i} \delta_{\mathbf{c}_i}(A), \text{ where } p_{\mathbf{X},i} = \frac{1}{n} \sum_{l=1}^{n} \mathbf{1}(\mathbf{X}_l \in \mathbf{c}_i), \text{ and } A \subset \mathbb{R}^d,$$

$$\hat{\mathbb{P}}_{\mathbf{Y}}(B) = \sum_{j=1}^{J} p_{\mathbf{Y},j} \delta_{\mathbf{c}_j}(B), B \subset \mathbb{R}^d.$$

The scalar $p_{\mathbf{X},i}$ (resp. $p_{\mathbf{Y},j}$) is the empirical weight around $\mathbf{c}_i$ (resp. $\mathbf{c}_j$) and induced from the sampling of $\mathbf{X}$ (resp. $\mathbf{Y}$). A
natural estimator of $\gamma \in \Gamma(\mathbb{P}_{\mathbf{X}}, \mathbb{P}_{\mathbf{Y}})$ can be written as

$$\hat{\gamma}(A \times B) = \sum_{i,j=1}^{I,J} \gamma_{i,j} \delta_{(\mathbf{c}_i, \mathbf{c}_j)}(A \times B).$$

The coefficients $\gamma_{ij}$ are the probabilities to transform $\mathbf{c}_i$ (i.e. a $\mathbf{x} \in \mathbf{c}_i$) to $\mathbf{c}_j$ (i.e. a $\mathbf{y} \in \mathbf{c}_j$). They are unknown, and they have to obey the marginal properties:

$$\sum_{j=1}^{J} \gamma_{ij} = p_{\mathbf{X},i}, \tag{2}$$

$$\sum_{i=1}^{I} \gamma_{ij} = p_{\mathbf{Y},j}. \tag{3}$$

Finally, the cost function defined in Eq. (1) can be approximated by

$$\hat{C}(\hat{\gamma}) = \sum_{i,j=1}^{I,J} \|\mathbf{c}_i - \mathbf{c}_j\|^2 \gamma_{ij} \tag{4}$$





Finding $\gamma_{ij}$, i.e. solving the problem defined by constraints of Eqs. (2-3) and minimization of Eq. (4), is called a *linear programming problem*. It can be solved (for example) by the network simplex algorithm (see, e.g. Bazaraa et al., 2009). We use the python implementation of Flamary and Courty (2017). To correct $\mathbf{X}$, we have to follow the plan of $\gamma_{ij}$. For a realization $\mathbf{X}_l$ of $\mathbf{X}$, we take the cell $\mathbf{c}_i$ that contains $\mathbf{X}_l$. Following $\hat{\gamma}$, $\mathbf{c}_i$ is moved to $\mathbf{c}_j$ with probability $\gamma_{ij}/p_{\mathbf{X},i}$ (applying Eq. (2), the sum over $j$ is 1). To find which $\mathbf{c}_j$, we draw it according to the conditional law $\hat{\gamma}_{\mathbf{X}_l} = (\gamma_{i1}, \ldots, \gamma_{iJ})/p_{\mathbf{X},i}$. Finally, we draw uniformly $\mathbf{y}$ in $\mathbf{c}_j$. This methodology is summarized in Algorithm 1, and we refer to it as *Optimal Transport Correction (OTC)*.

---

**Algorithm 1** Optimal Transport Correction (OTC)

---

**Require:** $(\mathbf{X}_1, \ldots, \mathbf{X}_n)$ a sample i.i.d. of the random variable $\mathbf{X}$

   $(\mathbf{Y}_1, \ldots, \mathbf{Y}_n)$ a sample i.i.d. of the random variable $\mathbf{Y}$

---

**Ensure:** $\mathbf{Z}_1, \ldots, \mathbf{Z}_n$ a sample i.i.d., correction of $(\mathbf{X}_1, \ldots, \mathbf{X}_n)$ with respect to the estimation of the law of $(\mathbf{Y}_1, \ldots, \mathbf{Y}_n)$

---

1: Estimate the law $\hat{\mathbb{P}}_{\mathbf{X}}$, from $(\mathbf{X}_1, \ldots, \mathbf{X}_n)$

2: Estimate the law $\hat{\mathbb{P}}_{\mathbf{Y}}$, from $(\mathbf{Y}_1, \ldots, \mathbf{Y}_n)$

3: Compute the optimal plan $\gamma_{ij}$ between $\hat{\mathbb{P}}_{\mathbf{X}}$ and $\hat{\mathbb{P}}_{\mathbf{Y}}$ (see, e.g. Flamary and Courty, 2017)

4: **for all** $\mathbf{X}_l$ **do**

5:    Find the cell $\mathbf{c}_i$ containing $\mathbf{X}_l$

6:    Construct the vector $\hat{\gamma}_{\mathbf{X}_l} = \left( \gamma_{i,1} \quad \ldots \quad \gamma_{i,J} \right)/p_{\mathbf{X},i}$ (The conditional law)

7:    Draw $j \in \{1, \ldots, J\}$ according to probability vector $\hat{\gamma}_{\mathbf{X}_l}$.

8:    Draw uniformly $\mathbf{Z}_l$ a realization of $\mathbf{Y}$ in cell $\mathbf{c}_j$.

9:    $\mathbf{Z}_l$ is a realization of $\mathbf{Z}$, corresponding to a correction of $\mathbf{X}_l$

10: **end for**

---

We propose an example, where $\mathbb{P}_{\mathbf{X}}$ is a standardized normal distribution and $\mathbb{P}_{\mathbf{Y}}$ a mixture of two normal distributions centered at 3 and $-3$, respectively. This example is univariate, and the corresponding plan is a histogram in dimension $d + d = 2$, see Fig. 1. The panels of Fig. 1 show the empirical histograms of $\hat{\mathbb{P}}_{\mathbf{X}}$ and $\hat{\mathbb{P}}_{\mathbf{Y}}$, with a bin size of 0.3. Under each histogram, the color scale represents the mass at each location. The central panel shows $\gamma_{ij}$ in $\mathbb{R}^2$. The black arrow coming from $\hat{\mathbb{P}}_{\mathbf{X}}$ splits the mass located in bins $[-0.3, 0)$ into eleven $\gamma_{ij}$. These $\gamma_{ij}$ are now moved into the segment $[-2, 1]$, following the grey arrows. To correct a point $x \in [-0.3, 0)$, we have to follow the black arrow, and draw a grey arrow, according to $\hat{\gamma}$.

Note that the traditional one-dimensional quantile mapping preserves the ordering of quantiles. In the multivariate case, this type of property can be viewed as the *Monge-Mather (1991) shortening principle* (see e.g. Villani, 2008, chap. 8). The idea is that the extremes of a multivariate distribution is moved to extremes, the boundary to the boundary, the level lines to level lines, etc.





## 2.4 Non stationary bias correction

Climate models offer a valuable tool to study future realistic climate trajectories. Climate model outputs of the present period need to be bias corrected with respect to current observations. Future climate simulations also need to be adjusted. However, no observation is available for the future and clear assumptions have to be made to correct simulations for future periods. Table 1

displays the basic framework of bias correction. Future unobserved data, say $\mathbf{Y}^1$, should be inferred from the current reference vector, $\mathbf{Y}^0$, and two numerical runs, one in the present, say $\mathbf{X}^0$, and one in the future, say $\mathbf{X}^1$. The period $0$ is called the *calibration period*, the period $1$ the *projection period*. In the univariate case, noting $F^i$ (resp. $G^i$) the CDF of $X^i$ (resp. $Y^i$), the CDF-$t$ (CDF transform) method of Michelangeli et al. (2009) assumes that

$$(G^1)^{-1} \circ G^0 = \mathcal{T}_{Y^0, Y^1} = \mathcal{T}_{X^0, X^1} = (F^1)^{-1} \circ F^0. \tag{5}$$

Recombining Eq. (5), the CDF of $Y^1$ is given by $G^1 = G^0 \circ (F^0)^{-1} \circ F^1$, and can be used to perform a quantile mapping correction. Here, the fundamental hypothesis $\mathcal{T}_{Y^0, Y^1} = \mathcal{T}_{X^0, X^1}$ means that the transfer functions to capture the temporal changes are identical in the model and observational worlds.

CDF-$t$ learns the change between $X^0$ and $X^1$, and transfers it to $Y^0$ to estimate $Y^1$. In the multivariate case, following CDF-

$t$, we want to learn the evolution (i.e. the change or the temporal evolution) between $\mathbf{X}^0$ and $\mathbf{X}^1$, and apply it to $\mathbf{Y}^0$. This generates $\mathbf{Y}^1$, and OTC can then be applied between $\mathbf{X}^1$ and $\mathbf{Y}^1$. Our definition of non-stationary bias correction assumes a transfer of the evolution of the model to observational world. But the evolution of observation can be different, and the resulting correction can be also different from observations. This methodology is justified because we want to keep the evolution of the model, even if the dynamic of the model is different of the dynamic of the observations.

Using OTC, we define two optimal plans. The optimal plan $\gamma$, between $\mathbf{X}^0$ and $\mathbf{Y}^0$, and the optimal plan $\varphi$, between $\mathbf{X}^0$ and $\mathbf{X}^1$. The law $\gamma$ is the *bias* between $\mathbf{X}^0$ and $\mathbf{Y}^0$, whereas $\varphi$ is the *evolution* between $\mathbf{X}^0$ and $\mathbf{X}^1$. Our goal is to move $\varphi$ *along* $\gamma$, defining a plan $\tilde{\varphi}$, to estimate $\mathbf{Y}^1$ as the evolution of $\mathbf{Y}^0$, i.e. $\mathbf{Y}^1 = \tilde{\varphi}(\mathbf{Y}^0)$. Then, we correct $\mathbf{X}^1$ with respect to $\mathbf{Y}^1 = \tilde{\varphi}(\mathbf{Y}^0)$, with OTC method. This is summarized in Fig. 2.

The estimation of $\tilde{\varphi}$ is performed in three steps:

1. transformation of $\varphi$ into a collection of vectors,

2. transfer of these vectors along $\gamma$ and

3. adaptation of these vectors to $\mathbf{Y}^0$.

To illustrate our methodology, Fig. 3 shows an example where the random variables $\mathbf{X}^0$, $\mathbf{X}^1$ and $\mathbf{Y}^0$ follow a bivariate Gaussian law. They are respectively centered at $(0,0)$, $(10,0)$ and $(0,10)$, with covariance matrices $4 \times \mathbf{Id}_2$, $\mathbf{Id}_2/4$ and $\mathbf{Id}_2/4$ (the matrix




$\mathbf{Id}_d$ is the $d$-dimensional identity matrix). Without loss of generality, we write the empirical distribution of $\mathbf{X}^0$, $\mathbf{X}^1$ and $\mathbf{Y}^0$ as a sum of Dirac masses,

$$\hat{\mathbb{P}}_{\mathbf{X}^0} = \sum_{i=1}^{I} p_{\mathbf{X}^0,i} \delta_{\mathbf{c}_i},$$

$$\hat{\mathbb{P}}_{\mathbf{Y}^0} = \sum_{j=1}^{J} p_{\mathbf{Y}^0,j} \delta_{\mathbf{c}_j},$$

$$\hat{\mathbb{P}}_{\mathbf{X}^1} = \sum_{k=1}^{K} p_{\mathbf{X}^1,k} \delta_{\mathbf{c}_k}.$$

**Step 1 : transformation of** $\varphi$**.** Using OTC method, $\varphi$ moves the bin $\mathbf{c}_i$ of $\hat{\mathbb{P}}_{\mathbf{X}^0}$ to the bin $\mathbf{c}_k$ of $\hat{\mathbb{P}}_{\mathbf{X}^1}$. The vector $\mathbf{v}_{ik} := \mathbf{c}_k - \mathbf{c}_i$ represents the evolution from $\mathbf{c}_i$ to $\mathbf{c}_k$ (i.e. the local evolution between $\mathbf{X}^0$ and $\mathbf{X}^1$). The collection of vectors $\mathbf{v}_{ik}$ is an estimation of the process between $\mathbf{X}^0$ and $\mathbf{X}^1$. In Fig. 3, the red arrow is an example of vector $\mathbf{v}_{ik}$.

**Step 2 : transfer along** $\gamma$**.** Using OTC method, $\gamma$ moves the bin $\mathbf{c}_i$ of $\hat{\mathbb{P}}_{\mathbf{X}^0}$ to the bin $\mathbf{c}_j$ of $\hat{\mathbb{P}}_{\mathbf{Y}^0}$. Thus, the estimation of $\tilde{\varphi}$ could be defined by the vector $\mathbf{v}_{ik}$ applied to $\mathbf{c}_j$, i.e. a realization of $\mathbf{Y}^1$ is given by $\mathbf{c}_j + \mathbf{v}_{ik}$. The grey arrow in Fig. 3a depicts this operation. But the $\mathbf{v}_{ik}$ can cross, and the correction is not coherent. This is due to normalizing issues and because the collection of vectors $\mathbf{v}_{ik}$ applied to $\mathbf{Y}^0$ do not define an optimal transport plan. The standard deviation decreases between $\mathbf{X}^0$ and $\mathbf{X}^1$, whereas it increases between $\mathbf{Y}^0$ and $\mathbf{Y}^1$. Consequently, we have to adapt the vectors $\mathbf{v}_{ik}$ to $\hat{\mathbb{P}}_{\mathbf{Y}^0}$.

**Step 3 : adaptation of** $\mathbf{v}_{ik}$**.** To solve this problem, we introduce a matrix factor $\mathbf{D}$, which rescales the collection of vectors $\mathbf{v}_{ik}$. In the univariate case, Bürger et al. (2011) proposed a factor $\sigma_{\mathbf{Y}^0}\sigma_{\mathbf{X}^0}^{-1}$, where $\sigma_{\bullet}$ is the standard deviation. The idea is to remove the scale of $\mathbf{X}^0$, and to replace it by the scale of $\mathbf{Y}^0$. Bárdossy and Pegram (2012); Cannon (2016) proposed a multivariate equivalent that uses the Cholesky decomposition of the covariance matrix. Noting $\Sigma$ the covariance matrix, and $\mathrm{Cho}(\Sigma)$ its

Cholesky decomposition, we multiply (in matrix sense) $\mathbf{v}_{ik}$ by the following matrix:

$$\mathbf{D} := \mathrm{Cho}(\Sigma_{\mathbf{Y}^0}) \cdot \mathrm{Cho}(\Sigma_{\mathbf{X}^0})^{-1}. \tag{6}$$

The Cholesky decomposition only exists if $\Sigma$ is symmetric, positive and definite. Some covariance matrices do not have this property, e.g. highly correlated random variables. In such a case, $\Sigma$ must be lightly perturbed to be definite (see, e.g. Higham, 1988; Knol and ten Berge, 1989). Furthermore, the Cholesky decomposition can be poorly estimated if the number of available data is too small compared to the dimension. Indeed, the inverse of a covariance matrix is highly biased. In this case, a pragmatic solution is to replace the matrix $\mathbf{D}$ by the diagonal matrix of standard deviation, i.e. $\mathbf{D} = \mathrm{diag}(\sigma_{\mathbf{Y}^0}\sigma_{\mathbf{X}^0}^{-1})$.





Finally, a realization of $\mathbf{Y}^1$ is given by $\mathbf{c}_j + \mathbf{D} \cdot \mathbf{v}_{ik}$. Figure 3b shows an estimation of $\mathbf{Y}^1$. Visually, the shape of $\mathbf{Y}^1$ appears coherent with the evolution between $\mathbf{X}^0$ and $\mathbf{X}^1$. The mean of $\mathbf{Y}^1$ is $(2.53, 10)$. The standard deviation between $\mathbf{X}^0$ and $\mathbf{X}^1$ is divided by $4$. The mean shift between $\mathbf{X}^0$ and $\mathbf{X}^1$ is $(10, 0)$. This shift of 10 units is correctly taken into account in the rescaling of $\mathbf{Y}^0$ by the standard deviation (equal to 4) between $\mathbf{X}^0$ and $\mathbf{X}^1$ :

$$\underbrace{(2.53, 10)}_{\mathbf{Y}^1 \text{ mean}} = \underbrace{(10, 0)}_{\text{mean shift between } \mathbf{X}^0 \text{ and } \mathbf{X}^1} / \underbrace{4}_{\text{Rescaling}} + \underbrace{(0, 10)}_{\mathbf{Y}^0 \text{ mean}} .$$

The value of the covariance matrix of $\mathbf{Y}^1$ is $\Sigma_{\mathbf{Y}^1} \simeq 0.018 \times \mathbf{Id}_2$. It is close of the expected value $(1/4)/16 \times \mathbf{Id}_2 \simeq 0.015 \times \mathbf{Id}_2$. The shift of 10 units of the model is not respected. It is interpreted as a correction of the bias into the evolution of the model. However, depending on the hypotheses desired by the user, the dOTC method can easily provide corrections whose mean evolutions and trends are in agreement with those given by the simulations to be corrected, like in the EDQM bias correction

method (Li et al., 2010). The complete method of correction is summarized in Algorithm 2. We refer to it by *dOTC* (dynamical Optimal Transport Correction).

We first propose to evaluate OTC and dOTC on an idealized case.

## 3 Bias correction on an idealized case

### 3.1 Model and methodology

To evaluate our bias correction method, we construct an idealized biased case, based on the Lorenz (1984) model. This three dimensional system is generated by the differential equations

$$\frac{\mathrm{d}\mathbf{x}}{\mathrm{d}t} = \begin{pmatrix} -x_2^2 - x_3^2 - (x_1 - \psi(t))/4 \\ x_1 x_2 - 4 x_1 x_3 - x_2 + 1 \\ x_1 x_3 + 4 x_1 x_2 - x_3 \end{pmatrix} . \tag{7}$$

The function $\psi(t)$ is a linear forcing proposed by Drótos et al. (2015). Classically, $\psi$ contains also a seasonal cycle (Lorenz, 1990), where the length of a "year" is fixed at $t = 73$ time units. Here we integrate this equation for the following forcing between 0 and $7 \times 73$ (i.e. 7 "years" of integration):

$$\psi(t) = 9.5 - 20 \frac{t - T}{T} \mathbf{1}_{\{t > T\}}, \quad T = 6 \times 73. \tag{8}$$

The integration is performed with a Runge-Kutta (order 4) scheme with a time step of size 0.005. All trajectories of the Lorenz
(1984) model converge on a unique subset of $\mathbb{R}^3$ (called an attractor), and remain trapped on it. According to Drótos et al.




---

**Algorithm 2** dynamical Optimal Transport Correction (dOTC)

---

**Require:** $(\mathbf{X}_1^0, \ldots, \mathbf{X}_n^0)$ a sample i.i.d. of the random variable $\mathbf{X}^0$

   $(\mathbf{X}_1^1, \ldots, \mathbf{X}_n^1)$ a sample i.i.d. of the random variable $\mathbf{X}^1$

   $(\mathbf{Y}_1^0, \ldots, \mathbf{Y}_n^0)$ a sample i.i.d. of the random variable $\mathbf{Y}^0$

---

**Ensure:** $(\mathbf{Z}_1^1, \ldots, \mathbf{Z}_n^1)$ a sample i.i.d. of the random variable $\mathbf{Z}^1$

---

1:  Estimate the law $\hat{\mathbb{P}}_{\mathbf{X}^0}$, from $(\mathbf{X}_1^0, \ldots, \mathbf{X}_n^0)$

2:  Estimate the law $\hat{\mathbb{P}}_{\mathbf{X}^1}$, from $(\mathbf{X}_1^1, \ldots, \mathbf{X}_n^1)$

3:  Estimate the law $\hat{\mathbb{P}}_{\mathbf{Y}^0}$, from $(\mathbf{Y}_1^0, \ldots, \mathbf{Y}_n^0)$

4:  Compute the optimal plan $\gamma_{ij}$ between $\hat{\mathbb{P}}_{\mathbf{X}^0}$ and $\hat{\mathbb{P}}_{\mathbf{Y}^0}$ (see, e.g. Flamary and Courty, 2017)

5:  Compute the optimal plan $\varphi_{ik}$ between $\hat{\mathbb{P}}_{\mathbf{X}^0}$ and $\hat{\mathbb{P}}_{\mathbf{X}^1}$

6:  Compute the Cholesky factor $\mathbf{D}$ between $(\mathbf{X}_1^0, \ldots, \mathbf{X}_n^0)$ and $(\mathbf{Y}_1^0, \ldots, \mathbf{Y}_n^0)$, given by Eq. (6).

7:  **for all $\mathbf{Y}_l^0$ do**

8:     Find the cell $\mathbf{c}_j$ containing $\mathbf{Y}_l^0$

9:     Using the plan $\gamma_{ij}$ (see Alg. 1), find a cell $\mathbf{c}_i$ of $\hat{\mathbb{P}}_{\mathbf{X}^0}$

10:     Using the plan $\varphi_{ik}$, find a cell $\mathbf{c}_k$ of $\hat{\mathbb{P}}_{\mathbf{X}^1}$

11:     Compute the vector $\mathbf{v}_{ik} := \mathbf{c}_k - \mathbf{c}_i$

12:     $\mathbf{Y}_l^1 = \mathbf{Y}_l^0 + \mathbf{D} \cdot \mathbf{v}_{ik}$ is a realization of $\mathbf{Y}^1$

13:  **end for**

14:  Estimate the law $\hat{\mathbb{P}}_{\mathbf{Y}^1}$, from $(\mathbf{Y}_1^1, \ldots, \mathbf{Y}_n^1)$

15:  Apply OTC (see Alg. 1) between $(\mathbf{X}_1^1, \ldots, \mathbf{X}_n^1)$ and $(\mathbf{Y}_1^1, \ldots, \mathbf{Y}_n^1)$ to generate $(\mathbf{Z}_1^1, \ldots, \mathbf{Z}_n^1)$

---

(2015), the five first "years" correspond to the time required to trap the trajectories.

One realization of random variable $\mathbf{Y}^0$ (resp. $\mathbf{Y}^1$) is the year 6 (resp. year 7). Each year contains $14600\,(= 73/0.005)$ elements. According to Eq. (8), the linear forcing is applied during the year 7. The non-stationarity is induced by the change between the two time periods.

We introduce a bias by multiplying each point of the trajectories by a triangular matrix $\mathbf{S}$, and add a vector $\mathbf{m}$, i.e. $\mathbf{X} = \mathbf{S}\mathbf{Y} + \mathbf{m}$. The addition changes the mean, whereas the multiplication alters the covariances. The matrix $\mathbf{S}$ is chosen empirically such that the covariance matrices of $\mathbf{X}^0$, $\mathbf{X}^1$, $\mathbf{Y}^0$ and $\mathbf{Y}^1$ differ. We fix:





$$\mathbf{S} = \begin{pmatrix} 1.22 & 0 & 0 \\ -0.41 & 1.04 & 0 \\ -0.41 & 0.56 & 0.52 \end{pmatrix}, \quad \mathbf{m} = \begin{pmatrix} 1 \\ 2 \\ 3 \end{pmatrix}.$$

The random variables $\mathbf{X}$ and $\mathbf{Y}$ are plotted in Fig. 4a and Fig. 4d. The blue (resp. red) curve of Fig. 4a is the trajectory of $\mathbf{Y}^0$ (resp. $\mathbf{X}^0$). The mean is largely altered. We estimate the covariance matrices as

$$\hat{\mathrm{Cov}}_{\mathbf{Y}^0} = \begin{pmatrix} 0.43 & -0.37 & -0.24 \\ -0.37 & 0.93 & 0.17 \\ -0.24 & 0.17 & 0.69 \end{pmatrix}, \quad \hat{\mathrm{Cov}}_{\mathbf{X}^0} = \begin{pmatrix} 0.64 & -0.68 & -0.62 \\ -0.68 & 1.39 & 1.0 \\ -0.62 & 1.0 & 0.92 \end{pmatrix}$$

Similarly to Fig. 4a, Fig. 4d depicts in blue $\mathbf{Y}^1$, and in red $\mathbf{X}^1$. The forcing of Eq. (8) has changed the properties of the trajectories, and they became chaotic. It is worthwhile to notice that the dynamic of $\mathbf{Y}$ is comparable to the one of $\mathbf{X}$. The covariance matrices are largely affected:

$$\hat{\mathrm{Cov}}_{\mathbf{Y}^1} = \begin{pmatrix} 0.27 & -0.09 & -0.14 \\ -0.09 & 0.81 & 0.08 \\ -0.14 & 0.08 & 0.73 \end{pmatrix}, \quad \hat{\mathrm{Cov}}_{\mathbf{X}^1} = \begin{pmatrix} 0.4 & -0.25 & -0.29 \\ -0.25 & 1.0 & 0.65 \\ -0.29 & 0.65 & 0.64 \end{pmatrix}$$

We estimate the empirical distributions $\mathbb{P}_{\mathbf{Y}^0}$, $\mathbb{P}_{\mathbf{Y}^1}$, $\mathbb{P}_{\mathbf{X}^0}$ and $\mathbb{P}_{\mathbf{X}^1}$ with a 3 dimensional histogram. We cut a large cube around

the trajectories into cells of size $0.2 \times 0.2 \times 0.2$. Then we count the number of points in each cell.

Finally, we evaluate the quality of the correction by comparing the covariance matrices of $\mathbf{Y}^0$ and $\mathbf{X}^0$, and the covariance matrices of $\mathbf{Y}^1$ and $\mathbf{X}^1$.

### 3.2   Correction of biased Lorenz (1984) model

We apply our method to correct $\mathbf{X}^0$ and $\mathbf{X}^1$. The random variable $\mathbf{X}^0$ is corrected with respect to $\mathbf{Y}^0$ and using the OTC method. The random variable $\mathbf{X}^1$ is corrected with respect to the estimation of $\mathbf{Y}^1$, coming from the dOTC method. The resulting random variables $\mathbf{Z}^0$ and $\mathbf{Z}^1$ are given in green on panels **b** and **e** of Fig. 4. We depict also on Fig. 4c and Fig. 4f a univariate correction with quantile mapping (resp. CDF-$t$) for the period 0 (resp. 1), generating the random variables $\mathbf{Q}^0$ (resp. $\mathbf{Q}^1$).

The correction $\mathbf{Z}^0$ is visually very similar to the reference in blue in Fig. 4a. The covariance matrix is almost perfectly reproduced




$$\hat{\mathrm{Cov}}_{\mathbf{Z}^0} = \begin{pmatrix} 0.42 & -0.36 & -0.24 \\ -0.36 & 0.93 & 0.17 \\ -0.24 & 0.17 & 0.69 \end{pmatrix}, \ \sup\left|\hat{\mathrm{Cov}}_{\mathbf{Z}^0} - \hat{\mathrm{Cov}}_{\mathbf{Y}^0}\right| = 0.004.$$

The correction $\mathbf{Z}^1$ is depicted in green in Fig. 4d. It is visually hard to compare to Fig. 4b, but we recognize $\mathbf{Y}^1$. The covariance matrix is correctly rectified

$$\hat{\mathrm{Cov}}_{\mathbf{Z}^1} = \begin{pmatrix} 0.26 & -0.11 & -0.11 \\ -0.11 & 0.82 & 0.08 \\ -0.11 & 0.08 & 0.71 \end{pmatrix}, \ \sup\left|\hat{\mathrm{Cov}}_{\mathbf{Z}^1} - \hat{\mathrm{Cov}}_{\mathbf{Y}^1}\right| = 0.03.$$

Finally, the cost of transformation (given by Eq. (1)) of $\mathbf{Z}^1$ into $\mathbf{Y}^1$ is 93% smaller than the cost between $\mathbf{Y}^1$ and $\mathbf{X}^1$, i.e. $\mathbb{P}_{\mathbf{Z}^1}$ is more similar to $\mathbb{P}_{\mathbf{Y}^1}$ than $\mathbb{P}_{\mathbf{X}^1}$. Furthermore, if we replace the Cholesky matrix of dOTC by the matrix of standard deviation, the maximum difference between covariance matrices increases to 0.22, but the cost is 85% smaller. Thus, using the standard deviation slightly degrades the correction. However, visually, it is very hard to distinguish the corrections with the Cholesky matrix or the standard deviation matrix. The figure corresponding to Fig. 4 with standard deviation matrix is given
in the supplementary material.

On the contrary, $\mathbf{Q}^0$ and $\mathbf{Q}^1$, depicted respectively in Fig. 4c and 4f, do not reproduce $\mathbf{Y}^0$ and $\mathbf{Y}^1$. Thus, the multivariate correction is largely better than the univariate correction. and $\mathbf{Y}^1$. It is confirmed by the covariance matrices, and $\mathbf{Y}^1$, which reproduce exactly the covariances of $\mathbf{X}^0$ and $\mathbf{X}^1$:

$$\hat{\mathrm{Cov}}_{\mathbf{Q}^0} = \begin{pmatrix} 0.42 & -0.42 & -0.42 \\ -0.42 & 0.95 & 0.68 \\ -0.42 & 0.68 & 0.69 \end{pmatrix}, \ \sup\left|\hat{\mathrm{Cov}}_{\mathbf{Q}^0} - \hat{\mathrm{Cov}}_{\mathbf{Y}^0}\right| = 0.51,$$

$$\hat{\mathrm{Cov}}_{\mathbf{Q}^1} = \begin{pmatrix} 0.13 & -0.1 & -0.14 \\ -0.1 & 0.59 & 0.39 \\ -0.14 & 0.39 & 0.46 \end{pmatrix}, \ \sup\left|\hat{\mathrm{Cov}}_{\mathbf{Q}^1} - \hat{\mathrm{Cov}}_{\mathbf{Y}^1}\right| = 0.31.$$

We have performed a tri-variate correction on a non linear system exhibiting non standard probability measures (i.e. non Gaussian, non exponential, etc.). In the stationary case, the OTC method works almost perfectly. In the non-stationary case, the dOTC method produces a probability distribution closed to the expected result. We propose now to apply OTC and dOTC
on climate model simulations.





## 4 Bias correction of an RCM simulation

### 4.1 Data

The dataset used as reference for the bias correction (BC) is the reanalysis "Systeme d'Analyse Fournissant des Renseignements Atmospheriques a la Neige" (SAFRAN, Vidal et al., 2010). SAFRAN is a hourly reanalysis over France between 1958

and present, with a horizontal resolution of $8$ km $\times$ $8$ km. Quintana-Seguí et al. (2008) claimed that daily mean of the Surface Atmospheric Temperature (tas) and Precipitation (pr) present no bias compared to observations from the climatological database of Météo-France. This justifies the use of SAFRAN as a reference.

We test our multivariate BC method on a simulation of the Weather Research and Forecast (WRF) atmospheric model (Ska-

marock et al., 2008) performed within the EURO-CORDEX initiative (Vautard et al., 2013; Jacob et al., 2014) with a $0.11° \times$ $0.11°$ horizontal resolution. The boundaries of the simulation were forced by a historical simulation of the Institut Pierre-Simon Laplace (IPSL) coupled model (Marti et al., 2010; Dufresne et al., 2013). This EURO-CORDEX historical simulation will be called "WRF" in the following.

SAFRAN and WRF data are re-mapped onto the same grid, with a spatial resolution of $0.11° \times 0.11°$ (i.e. $\sim 12$km $\times$ $12$km). The closest neighborhood method is used. We only keep the land region comprised in $1.8 - 7.85° \ E \times 41.8 - 45.2° \ N$ i.e. covering the south-east of France. This region is characterized by a complex topography, which creates a strong spatial heterogeneity, especially for precipitation. For the present application, we extract $12$ grid points regularly spaced, see Fig. 5a, with a one to one spatial correspondence between SAFRAN and WRF.

In both datasets, we will consider daily surface air temperatures and precipitation. The goal of this section is to correct the bias in tas and pr in the WRF data with respect to SAFRAN.

### 4.2 Cross-validation protocol

We focus on the daily time scale over the 1970–2000 period. We correct the warm season (May–September). The analysis and

conclusions are available for the cold season, and the corresponding figure (ie. Fig.5) is given in the supplementary material. We split that period into two sub-periods, 1970-1985 (2295 days), and 1985-2000 (2295 days) to perform a cross-validation. The SAFRAN (resp. WRF) values over the first time period correspond to the random variable $\mathbf{Y}^0$ (resp. $\mathbf{X}^0$), and is called the *calibration period*. The SAFRAN (resp. WRF) values over the second time period correspond to $\mathbf{Y}^1$ (resp. $\mathbf{X}^1$), and is called the *projection period*. SAFRAN during 1985-2000 (i.e. $\mathbf{Y}^1$) is assumed to be unknown, and is used for cross-validation.


We perform two bias corrections: univariate and $24$-variate (12 grid points and 2 variables).

1. For univariate correction, quantile mapping is used for the calibration period, and CDF-$t$ for the projection period.




2. For 24-variate correction, OTC is used for the calibration period, and dOTC for the projection period. The spatial structure and the dependence between the two variables are used. Due to the dimension, the Cholesky matrix is poorly estimated. We replace it by the matrix of standard deviation in the rescaling step.

We estimate the empirical distributions by computing histograms with bins of size $0.1$ in each dimension. Furthermore, CDF-$t$
and dOTC can shift close to $0$ values to negative values for precipitation. Thus, negative precipitation values are replaced by $0$ after correction. We test the quality of the correction by plotting the evolution of the mean, of the standard deviation, and the spatial and inter-variables covariance, i.e. the difference between projection and calibration period. These indicators are summarized in Fig. 5. During calibration period, the goal is that the probability distribution of correction of the WRF simulation is the probability distribution of SAFRAN. By construction of OTC, the correction is almost perfect, and we focus on projection
period. In projection period, the goal is that the evolution of corrections is close to the evolution of WRF simulation.

## 4.3   Evolution analysis

As we have seen in previous section, the correction of $\mathbf{X}^1$ and $\mathbf{Y}^1$ are identical only if the evolution of SAFRAN is identical to the evolution of WRF. To analyze the evolution of WRF, SAFRAN and the corrections, we compute the difference of statistical indicators between the projection and the calibration period at each grid points. The indicators are the mean (Fig. 5b,f), the
variance (Fig. 5c,g), the covariance between pr and tas (Fig. 5e) and the spatial covariance for each variables (Fig. 5d,h).

The $x$ axis of Figs. 5a-h is the evolution of the correction (i.e. $\mathbb{E}(\mathbf{Z}^1) - \mathbb{E}(\mathbf{Z}^0),\ldots$). The $y$ axis of Figs. 5a-h is the evolution of WRF in red (i.e. $\mathbb{E}(\mathbf{X}^1) - \mathbb{E}(\mathbf{X}^0),\ldots$), and the evolution of SAFRAN in blue (i.e. $\mathbb{E}(\mathbf{Y}^1) - \mathbb{E}(\mathbf{Y}^0),\ldots$). Furthermore, the red line is the linear regression between the evolution of the 24-variate correction and the evolution of WRF. The correlation
($r$-value), $p$-value and standard error of each linear regression are summarized in Tab. 2.

The linear regression between evolution of 24-variate correction and evolution of WRF (red line) shows a strong statistical link for all statistical indicators. The evolution of the mean is almost perfectly reproduced for the two variables ($r$-values is at least equal to $0.98$, with a maximal $p$-value at $10^{-9}$). The evolution of variance of WRF is also reproduced, the linear regression
being significative (maximal $p$-value is $5 \times 10^{-2}$).

The evolution of dependence structure is given by the evolution of spatial and inter-variables covariance. The minimal $r$-value for linear regression is equal to $0.59$ with a maximal $p$-value equal to $2 \times 10^{-3}$. This means that dOTC reproduces the evolution of WRF between calibration and projection period. Because the calibration period is perfectly corrected, the correction during
projection period appears as the evolution of WRF, applied to SAFRAN.

A linear regression, the Spearman rank correlation between the evolution of SAFRAN, and the evolution of the correction with WRF do not show a significant statistical link (not shown). We conclude that the evolution of WRF is different of the evolution





of SAFRAN. This indicates it is not possible to reproduce SAFRAN during projection period using dOTC and WRF.

The correction with CDF-$t$ appears to be satisfactory for the temperatures, and very similar to the correction with dOTC. But for the precipitation, the structure is not coherent with WRF or SAFRAN. This dissimilarity is due to the difference between
the probability distribution of temperatures (quasi-Gaussian) and precipitations (exponential/Gamma laws).

We conclude that the evolution of the 24-variate correction with dOTC between calibration and projection period is close to the evolution of WRF. Furthermore, the evolution of SAFRAN is very different from the evolution of WRF.

## 5   Conclusions

We have developed a new method for multivariate bias correction, generalizing the quantile mapping in the multivariate case. To do so, we have developed a new theoretical framework to understand any bias correction (BC) method: any BC method is here characterized by a joint law between the biased dataset and the correction. This joint probability distribution is estimated based on optimal transport techniques, and the BC method is then refer to as "Optimal Transport Correction" (OTC). A definition of non-stationary bias correction is also proposed: the evolution of the model is learned, and transfered to the reference world. An
extension of OTC called dynamical OTC (dOTC) has been developed to account for temporal non-stationarities.
OTC and dOTC methods have been tested on an idealized 3 dimensional case based on Lorenz (1984) time-dependent attractors, which induced changes in the correlation between variables. The bias correction appeared to perform very well in those idealized experiments.

Then, 12 grid points of a WRF simulation have been corrected with respect to SAFRAN reanalyses for precipitation and temperature in Southern France. A 24-variate correction was performed. The correction in stationary context was almost perfect. In the non-stationary case, the evolutions of WRF and SAFRAN were different, and, as expected, the correction with dOTC differed from SAFRAN. However, the correction presented a multidimensional evolution similar to that of WRF. We can therefore conclude that the correction is consistent with the definition proposed for the non-stationary case.

This is consistent with the results of Maraun et al. (2017): the fundamental errors of a model are not corrected, but transferred to the world of observations. The dOTC method preserves the signal of climate change inferred from the model simulations. As suggested by Maraun and Widmann (2018) our cross-validation method does not compare the correction to the observations on the validation period, which can produce false positive or true negative due to internal variability of model or observations,
but assesses whether the statistical evolution of the model is kept.

Furthermore, although the number of available data is very small compared to the dimension (2295 days and 24 dimensions), the OTC and dOTC performed a correction without numerical problems, and, moreover, only in a few minutes on a personal

computer.

As a perspective of improvement of the method, we note that the optimal plan can only be used to correct data points that are already known. If a new data point is obtained, and alters the estimate of the probability density function, then the plan

needs to be recomputed. However, such a situation is relatively rare in bias correction. Indeed, the corrections usually have to be performed on climate model simulations that cover many years and decades. This means that the whole time series are available at once and are not continuously updated. One possibility would be to "smooth" the optimal plan that, thus, could be applied to new points whithout recalculating the plan. Finally, a promising application of this method is the post-processing of operational forecasts. In such a case, the question of internal variability (Maraun et al., 2017) would not affect the bias

correction procedure as climate dynamics is consistently represented between the model and observations.

*Code availability.*   OTC and dOTC are implemented in two packages: ARyga (R) and Apyga (python3). These packages are available upon request.

*Data availability.*   TEXT

*Code and data availability.*   TEXT

*Author contributions.*   YR performed the analyses. The experiments were co-designed by YR and MV. All the authors contributed to writing the manuscript

*Competing interests.*   The authors declare no competing interest.

*Acknowledgements.*   This work was supported by ERC Grant No. 338965-A2C2.





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





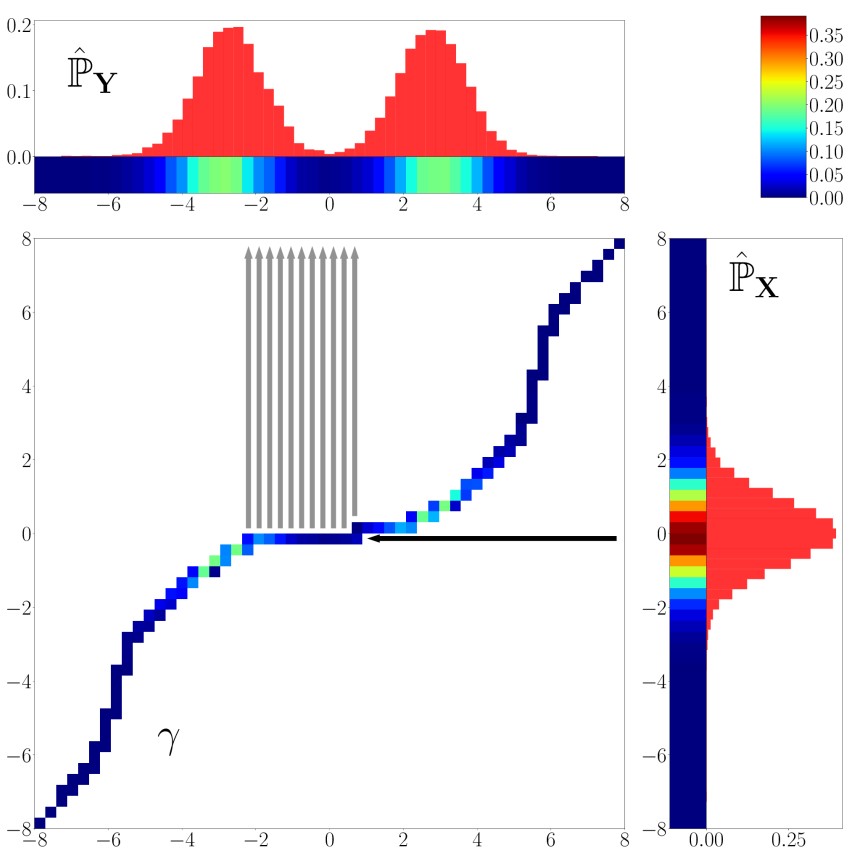

**Figure 1.** The right panel of $\hat{\mathbb{P}}_{\mathbf{X}}$ is the histogram of the standardized normal law. The upper panel of $\hat{\mathbb{P}}_{\mathbf{Y}}$ is the histogram of a mixture of two normal laws, centered in $-3$ and $3$. The size of bins is $0.3$. Under each histogram, the color scale represents the mass at each location. The central panel is the optimal joint law, minimizing the Eq. (4). The black and grey arrow represents the correction with Alg. 1.

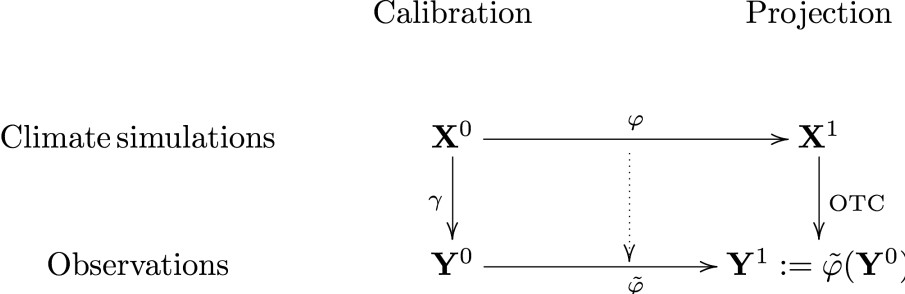

**Figure 2.** Estimation of the unobserved random variable $\mathbf{Y}^1$. The random variables $\mathbf{X}^0$, $\mathbf{X}^1$ and $\mathbf{Y}^0$ are known. The plans $\gamma$ and $\varphi$ are the optimal joint laws in the sense of equations (2-4). $\tilde{\varphi}$ is the evolution of $\mathbf{Y}^0$ estimated from $\gamma$ and $\varphi$. OTC is used to correct $\mathbf{X}^1$ with respect to the estimation of $\mathbf{Y}^1$.





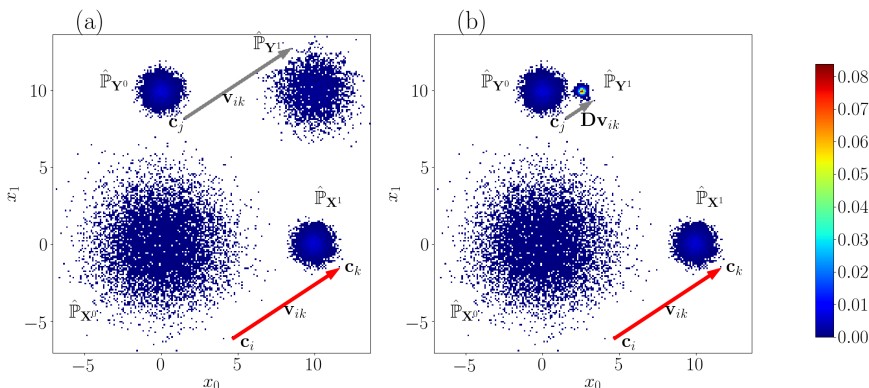

**Figure 3.** Bivariate histogram with bin size equal to 0.1. On each Panel we have: a Gaussian law centered in $(0,0)$ with covariance $4\mathbf{Id}_2$ ($\hat{\mathbb{P}}_{\mathbf{X}^0}$), a Gaussian law centered in $(10,0)$ with covariance $1/4\mathbf{Id}_2$ ($\hat{\mathbb{P}}_{\mathbf{X}^1}$) and a Gaussian law centered in $(0,10)$ with covariance $1/4\mathbf{Id}_2$ ($\hat{\mathbb{P}}_{\mathbf{X}^1}$). The red arrow is the local evolution between $\hat{\mathbb{P}}_{\mathbf{X}^0}$ and $\hat{\mathbb{P}}_{\mathbf{X}^1}$. (a) The probability distribution $\hat{\mathbb{P}}_{\mathbf{Y}^1}$ is the correction with OTC-$t$ and $\mathbf{D} = \mathrm{Id}_2$. The grey arrow is the estimation of the evolution of $\hat{\mathbb{P}}_{\mathbf{Y}^0}$. (b) The probability distribution $\hat{\mathbb{P}}_{\mathbf{Y}^1}$ is the correction with dOTC and $\mathbf{D}$ given by Eq. (6). The grey arrow is the estimation of the evolution of $\hat{\mathbb{P}}_{\mathbf{Y}^0}$.





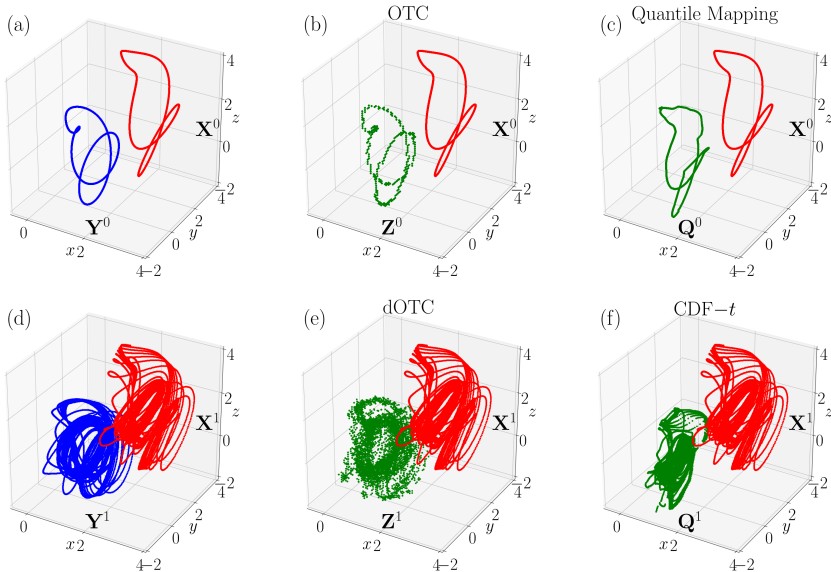

**Figure 4.** Random variables generated by Lorenz (1984) model, OTC, dOTC, quantile mapping and CDF-$t$. (a) Biased random variable $\mathbf{X}^0$ (red) and references $\mathbf{Y}^0$ (blue) for time period 0. (b) Biased random variable $\mathbf{X}^0$ (red) and correction $\mathbf{Z}^0$ with OTC (green). (c) Biased random variable $\mathbf{X}^0$ (red) and correction $\mathbf{Q}^0$ with quantile mapping (green). (d) Biased random variable $\mathbf{X}^1$ (red) and references $\mathbf{Y}^1$ (blue) for time period 1. (e) Biased random variable $\mathbf{X}^1$ (red) and correction $\mathbf{Z}^1$ with dOTC (green). (f) Biased random variable $\mathbf{X}^1$ (red) and correction $\mathbf{Q}^1$ with CDF-$t$ (green).





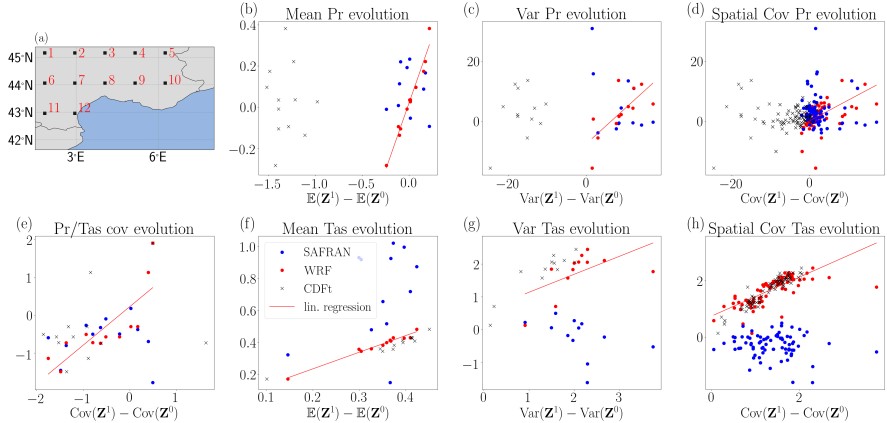

**Figure 5.** (a) Map of the south east of the France. The 12 black squares are the locations where corrections are performed. (b-h) The $x$ axis of panels a-h is the evolution of the correction with dOTC. The $y$ axis of panels a-h is the evolution of WRF in red and the evolution of SAFRAN in blue. The red line is the linear regression between the evolution of correction and the evolution of WRF. The black cross marker are the scatter plot between the evolution of correction with CDF-$t$ and evolution of WRF. (b) Evolution of mean precipitation, i.e. difference between the projection period and the calibration period. (c) Evolution of variance of precipitation. (d) Evolution of spatial covariance of precipitation. (e) Evolution of covariance between precipitation and temperatures. (f) Evolution of mean temperatures. (g) Evolution of variance of temperatures. (h) Evolution of spatial covariance of temperatures.





**Table 1.** Representation of bias correction in context of climate change.

|  | Present | Future |
|---|---|---|
| Numerical model | $\mathbf{X}^0$ | $\mathbf{X}^1$ |
| Observations | $\mathbf{Y}^0$ | unknown ($\mathbf{Y}^1$) |

**Table 2.** $r$-value, $p$-value and standard error of linear regression between evolution of correction and evolution of WRF.

|  | $r$-value | $p$-value | standard error |
|---|---|---|---|
| Mean evolution Pr | 0.98 | $10^{-8}$ | 0.08 |
| Mean evolution Tas | 0.99 | $10^{-9}$ | 0.05 |
| Variance evolution Pr | 0.71 | $10^{-2}$ | 0.37 |
| Variance evolution Tas | 0.57 | $5 \times 10^{-2}$ | 0.25 |
| Covariance Pr/Tas evolution | 0.81 | $2 \times 10^{-3}$ | 0.24 |
| Spatial covariance Pr | 0.59 | $10^{-14}$ | 0.08 |
| Spatial covariance Tas | 0.76 | $10^{-29}$ | 0.05 |