# Peer review of "Multivariate stochastic bias corrections with optimal transport"

_Hydrology and Earth System Sciences, 2018_

## Referee Comment (RC1) · Anonymous Referee #1 · 13 Aug 2018

**Manuscript:** Multivariate stochastic bias corrections with optimal transport

**Major remarks**

The authors introduce a new method of stochastic bias correction, which is based on optimal transport. The new method can be used for multivariate cases, and it is also extended for non-stationary applications. When showing results, the methods yields reasonable results. As the methods is also supposed to be fast, it is a valuable contribution for bias correction applications where more than one variable shall be corrected and that aim at keeping interdependence structures between the corrected variables.

Unfortunately, I can judge neither whether the method is soundly derived nor what the method is really doing. Section 2 (and partially also section 3) comprises a heavy formalism and is not human readable without a profound statistical background, which, I assume, most HESS readers like me do not have. Even though I am familiar with bias correction methods, e.g. based on quantile mapping, I got lost in section 2. On the one hand, by using this heavy statistical formalism, nomenclature and terms, the paper may be better suited for a mathematically or statistically oriented journal. On the other hand, the new method is interesting for the hydrological and climate impact modelling communities, so that I suggest a major rewriting of this section. This should be done in a way by using a more descriptive approach, which can be understood by readers who are not experts in statistics. This approach may include some simple examples to explain specific terms of the method whose use is unavoidable. These examples may comprise demonstrating explanations for a case where precipitation and temperature are corrected at the same time (In this way linking to the application of the method in Section 4.). Some more technical parts, which are necessary for the mathematical derivation of the method, may be put into the appendix to support also those readers who are interested in the mathematical details.

I also miss a discussion of the method and its results in comparison to other studies that considered the joint correction of precipitation and temperature, e.g. Piani and Härter (2012) or Räty et al. (2018).

Piani, C.; Haerter, J.O. Two dimensional bias correction of temperature and precipitation copulas in climate models. Geophys. Res. Lett. 2012, 39.

Räty, O.; Räisänen, J.; Bosshard, T.; Donnelly, C. Intercomparison of Univariate and Joint Bias Correction Methods in Changing Climate From a Hydrological Perspective. Climate 2018, 6, 33.

In case (see above), major revisions will be conducted, the paper may be accepted for publication.

**Minor remark**

p. 13 - line 16
It is written:
"The closest neighborhood method is used."
I assume you mean the "nearest neighbor interpolation". Please rewrite accordingly.

---

## Referee Comment (RC2) · M Muskulus (Referee) · 3 Sep 2018

**Synposis**

The manuscript studies bias-correction methods for climate models (that are here considered as dynamical systems). A new method based on optimal transport theory is suggested: given results from two models X and Y and considering these as probability distributions, a joint probability law is determined from optimal transport theory that couples the two distributions for X and Y in a certain optimal sense (i.e., least work in transforming X into Y). The ensuing joint distribution can then be used to obtain stochastic corrections for data samples, by sampling from the conditional distribution of Y, given X (or vice versa) - which is an interesting idea. The authors then continue

to suggest corrections for the case of observing two models at two different points in time. Here they propose to estimate the optimal transport from Y0 to X0 and from X0 to X1. Together, these two joint distributions are used to obtain values of Y1 from Y0, by adding random realizations of the differences between X0 and X1 to Y0, scaled by the covariance matrix between X0 and Y0. Simulations with a nonstationary, perturbed variant of the Lorenz model show the potential for reconstruction of the (known) distribution of Y1. Results comparing two different climate models are somewhat less satisfactory and show that time evolution in both models seems to differ in some important aspects that are not captured too well by the method then. However, the correction method at single time instances still works well.

**Assessment**:

The manuscript is well written and the topic is suitable for HESS. The explanation of the proposed methods is well executed, being correct and relevant, while being concise and leaving out unnecessary mathematical details. All in all a manuscript that I enjoyed reading. It is actually quite thought-provoking, since the correction method proposed for the "nonstationary" case could also be done differently - maybe it would be worth to investigate these other options also? I can recommend the manuscript for publication, but I would like to invite the authors to comment on a few issues (see detailed comments below), mostly in order to further increase the relevance.

**Detailed comments**:

1. Equation 1: The authors use the quadratic Wasserstein distance. This has a number of nice theoretical properties, but in statistics the L1 Wasserstein distance (i.e. without the square, then known as the "Kantorovich-Rubinstein" distance) would be a more robust choice — although potentially loosing uniqueness of the solution then — and might be considered. Please add a few words about the choice of the distance here.

2. The way the "nonstationary" case is addressed is very interesting, but also somewhat controversial. Both the CDF-t method and the authors' work is based on assuming that time evolution is somehow the same for the two models/systems considered, i.e., that

$$T_{Y^1,Y^0} = T_{X^0,X^1}.$$

This might be justified from a dynamical point of view, but from a statistical (or data science) point of view it seems more reasonable to actually consider that the bias between the two systems remains the same, i.e., that

$$T_{Y^1,X^1} = T_{Y^0,X^0}$$

The transformation would then be the opposite, $G^1 = F^1 \cdot (F^0)^{-1} \cdot G^0$ instead of $G^1 = G^0 \cdot (F^0)^{-1} \cdot F^1$ as for the above. Has nobody considered this so far in the literature? Why not? The paper would benefit a lot from a (short) discussion of this second possibility! (and maybe even a few results with it)

3. page 7, line 18: Related to the previous item, "because we want to keep the evolution of the model" is a somewhat unscientific statement. Why do you want to assume that the time evolution is the same - what are the reasons that make this a suitable assumption here, especially for complex climate models?

4. page 8, line 14: "whereas it increases between ..." - It does, but only here for this example, not in general. Please mention this, to avoid confusion.

5. page 8, step 3: The proposed adaption seems somewhat unnatural to me. Looking at Figure 3, I would think that Figure 3a is a more appropriate reconstruction than Figure 3b, since it captures the important fact that the uncertainty about the values increases when transferring the assume dynamical evolution. This seems actually desirable! But of course it all depends on the goal here. Please comment!

6. Finally, it could be nice to discuss in the conclusions the relationship with copulas - which are functions that capture the dependence structure between two random variables X, Y, as does the transport plan here - so there is an underlying "optimal transport copula". Mentioning this connection could maybe make the work presented here interesting for a larger readership.

**Minor comments**

- page 3, line 13: Just a comment: the nomenclature is a bit strange (this seems to have historic roots in this field, so is not the authors' fault), the name "transfer function" does not seem a good choice as it means something quite different in dynamics. It would be more appropriate to simply call this a "map".

- page 3, line 15: maybe add "deterministic" before "transfer function"?

- page 12, line 19: "close" instead of "closed"

- page 8, line 22-23: A matrix is not "definite", so you probably mean "positive-definite" here in both cases?

- page 14, line 25: "significant" instead of "significative"

- page 14: The discussion of the results shown in Figure 5 is quite dense, it could benefit from a few more words?

---

## Author Comment (AC1) · 14 Nov 2018

**1   Major remarks**

*The authors introduce a new method of stochastic bias correction, which is based on optimal transport. The new method can be used for multivariate cases, and it is also extended for non- stationary applications. When showing results, the methods yields reasonable results. As the methods is also supposed to be fast, it is a valuable contribution for bias correction applications where more than one variable shall be corrected and that aim at keeping interdependence structures between the corrected variables. Unfortunately, I can judge neither whether the method is soundly derived nor what the method is really doing. Section 2 (and partially also section 3) comprises a*

*heavy formalism and is not human readable without a profound statistical background, which, I assume, most HESS readers like me do not have. Even though I am familiar with bias correction methods, e.g. based on quantile mapping, I got lost in section 2. On the one hand, by using this heavy statistical formalism, nomenclature and terms, the paper may be better suited for a mathematically or statistically oriented journal. On the other hand, the new method is interesting for the hydrological and climate impact modelling communities, so that I suggest a major rewriting of this section. This should be done in a way by using a more descriptive approach, which can be understood by readers who are not experts in statistics. This approach may include some simple examples to explain specific terms of the method whose use is unavoidable. These examples may comprise demonstrating explanations for a case where precipitation and temperature are corrected at the same time (In this way linking to the application of the method in Section 4.). Some more technical parts, which are necessary for the mathematical derivation of the method, may be put into the appendix to support also those readers who are interested in the mathematical details. I also miss a discussion of the method and its results in comparison to other studies that considered the joint correction of precipitation and temperature, e.g. Piani and Härter (2012) or Räty et al. (2018).*

- Piani, C.; Haerter, J.O. Two dimensional bias correction of temperature and precipitation copulas in climate models. Geophys. Res. Lett. 2012, 39.

- Räty, O.; Räisänen, J.; Bosshard, T.; Donnelly, C. Intercomparison of Univariate and Joint Bias Correction Methods in Changing Climate From a Hydrological Perspective. Climate 2018, 6, 33.

*In case (see above), major revisions will be conducted, the paper may be accepted for publication.*

**1.0.1 Response**

It is always difficult to find the right balance between statistical formalism and pedagogical aspects. While Reviewer 2 enjoyed reading our manuscript for its concise and light mathematical contents, it is true that the lack of a simple example limited the accessibility to readers with a more applied background. In this context, we have decided to add an illustrative example, see the updated section 2.1 and the new Figure 1, that, hopefully, provides the basic visual elements to understand 1d quantile mapping as an optimal transport problem. We hope this change, while keeping a clear formalism, will allow more readers to follow the main idea at hand.

**1.0.2 Modification**

A new sub-section 2.1 has been added. This explain how our bias correction method is built on an example similar to temperature, and the meaning of the notations. Furthermore, we have added a discussion on how our method compares to others in the conclusion.

**2 Minor remark**

**2.1 p. 13 - line 16**

*It is written: "The closest neighborhood method is used." I assume you mean the "nearest neighbor interpolation". Please rewrite accordingly.*

**2.1.1 Response**

We agree with the reviewer

**2.1.2 Modification**

The change has been done.
* * *

---

## Author Comment (AC2) · 14 Nov 2018

**1   Synopsis**

*The manuscript studies bias-correction methods for climate models (that are here considered as dynamical systems). A new method based on optimal transport theory is suggested: given results from two models $\mathbf{X}$ and $\mathbf{Y}$ and considering these as probability distributions, a joint probability law is determined from optimal transport theory that couples the two distributions for $\mathbf{X}$ and $\mathbf{Y}$ in a certain optimal sense (i.e., least work in transforming $\mathbf{X}$ into $\mathbf{Y}$). The ensuing joint distribution can then be used to obtain stochastic corrections for data samples, by sampling from the conditional distribution of $\mathbf{Y}$, given $\mathbf{X}$ (or vice versa) - which is an interesting idea. The authors*

[Figure]

*then continue to suggest corrections for the case of observing two models at two different points in time. Here they propose to estimate the optimal transport from $\mathbf{Y}^0$ to $\mathbf{X}^0$ and from $\mathbf{X}^0$ to $\mathbf{X}^1$. Together, these two joint distributions are used to obtain values of $\mathbf{Y}^1$ from $\mathbf{Y}^0$, by adding random realizations of the differences between $\mathbf{X}^0$ and $\mathbf{X}^1$ to $\mathbf{Y}^0$, scaled by the covariance matrix between $\mathbf{X}^0$ and $\mathbf{Y}^0$. Simulations with a non-stationary, perturbed variant of the Lorenz model show the potential for reconstruction of the (known) distribution of $\mathbf{Y}^1$. Results comparing two different climate models are somewhat less satisfactory and show that time evolution in both models seems to differ in some important aspects that are not captured too well by the method then. However, the correction method at single time instances still works well.*

**2 Assessment**

*The manuscript is well written and the topic is suitable for HESS. The explanation of the proposed methods is well executed, being correct and relevant, while being concise and leaving out unnecessary mathematical details. All in all a manuscript that I enjoyed reading. It is actually quite thought-provoking, since the correction method proposed for the " non-stationary" case could also be done differently - maybe it would be worth to investigate these other options also? I can recommend the manuscript for publication, but I would like to invite the authors to comment on a few issues (see detailed comments below), mostly in order to further increase the relevance.*

Thank you for this comment. This is right: the non-stationarity can be done differently but some preliminary tests performed on another approach gave disappointing results (see our answer to comment about Section 2.4). We propose this approach because we can define properly the dynamic of the model, and the idea "to transfer the dynamic

to observations" is justified. The reverse hypothesis (transfer the bias) is more problematic, because we do not really understand "what" the bias is, and assuming the bias is the same at two time period is not justified for us. More investigation (in progress) is necessary to understand first how the dynamics of the model is preserved (or not) using more robust indicators coming from dynamical system theory (see, e.g., Freitas, (2010,2012), Lucarini (2012) ), and secondly to define what the "bias" is.

- Freitas, A. C. M., Freitas, J. M. and Todd, M. (2010). "Hitting time statistics and extreme value theory". In : Probability Theory and Related Fields 147.3, p. 675–710. doi : 10.1007/s00440-009-0221-y.

- Freitas, A. C. M., Freitas, J. M. and Todd, M. (2012). "The extremal index, hitting time statistics and periodicity". In : Advances in Mathematics 231.5, p. 2626–2665. doi : https://doi.org/10.1016/j.aim.2012.07. 029.

- Lucarini, V., Faranda, D. and Turchetti, G. and Vaienti, S. (2012). "Extreme value theory for singular measures". In : Chaos : An Interdisciplinary Journal of Nonlinear Science 22.2, p. 023135. doi : 10.1063/1.4718935.

**3   Detailed comments**

**3.1   Equation 1**

*The authors use the quadratic Wasserstein distance. This has a number of nice theoretical properties, but in statistics the $L^1$ Wasserstein distance (i.e. without the square, then known as the " Kantorovich-Rubinstein" distance) would be a more robust choice — although potentially loosing uniqueness of the solution then — and might be considered. Please add a few words about the choice of the distance here.*

**3.1.1 Response**

We have considered the use of the $L^1$ Wasserstein distance. In the case of the Lorenz 84 attractor (Section 3 of manuscript), using the $L^1$ or the $L^2$ distance does not change the correction for OTC and dOTC method. But for the example with three bivariate Gaussian (dOTC correction, see Section 2.5), the $L^1$ distance gives results that are not satisfactory (see Fig. **??**, same experiment as Section 2.5, with $L^1$ distance). In the $y$-axis, the standard deviation is equal to $0.61$, whereas $0.125$ is expected. So we have chosen the $L^2$ distance.

**3.1.2 Modification**

A short explanation has been added p. 6, l. 13-14.

**3.2 Section 2.4**

*The way the " non-stationary" case is addressed is very interesting, but also somewhat controversial. Both the CDF-t method and the authors' work is based on assuming that time evolution is somehow the same for the two models/systems considered, i.e., that*

$$\mathcal{T}_{Y^1,Y^0} = \mathcal{T}_{X^1,X^0}$$

*This might be justified from a dynamical point of view, but from a statistical (or data science) point of view it seems more reasonable to actually consider that the bias between the two systems remains the same, i.e., that*

$$\mathcal{T}_{Y^1,X^1} = \mathcal{T}_{Y^0,X^0}$$

*The transformation would then be the opposite, $G^1 = F^1 \circ (F^0)^{-1} \circ G^0$ instead of $G^1 = G^0 \circ (F^0)^{-1} \circ F^1$ as for the above. Has nobody considered this so far in the literature? Why not? The paper would benefit a lot from a (short) discussion of this second possibility! (and maybe even a few results with it)*

**3.2.1 Response**

As far as we know, no one in the literature has considered the opposite relationship. Almost all bias correction methods assume that the "bias" is constant, but this notion is not clearly defined. For example, with our formalism, the Quantile Matching method assumes in practice the stationnarity (i.e. $\mathbb{P}_{\mathbf{Y}^0} = \mathbb{P}_{\mathbf{Y}^1}$), and CDF-$t$ a transfer of the dynamic. At the beginning of our work, we wanted to use both hypotheses simultaneously (i.e. $\mathcal{T}_{Y^1,Y^0} = \mathcal{T}_{X^1,X^0}$ and $\mathcal{T}_{Y^1,X^1} = \mathcal{T}_{Y^0,X^0}$). In this case, the only solutions are trivial: $\mathbb{P}_{X^0} = \mathbb{P}_{X^1}$ or $\mathbb{P}_{X^0} = \mathbb{P}_{Y^0}$ or $\mathbb{P}_{X^0} = \mathbb{P}_{Y^1}$. We therefore concluded that it was necessary to choose one of the two hypotheses but it is impossible to have both in practice. We retained the hypothesis of the evolution that made it possible to reconstruct the Lorenz84.

**3.2.2 Modification**

We change the paragraph:

*"Our definition of non-stationary bias correction assumes a transfer of the evolution of the model to observational world. But the evolution of observation can be different, and the resulting correction can be also different from observations. This methodology is*

*justified because we want to keep the evolution of the model, even if the dynamic of the model is different of the dynamic of the observations."*

For (p. 8, l. 11-17 and p.9, l. 1-2):

*"Note also that the reverse hypothesis $\mathcal{T}_{Y^1,X^1} = \mathcal{T}_{Y^0,X^0}$ could be considered, meaning that the bias is learned, and transferred along the dynamic. In this case, the correction of example given in Section 3 does not correspond to the reference (not shown), so we rejected this assumption. Thus, our definition of non-stationary bias correction assumes a transfer of the evolution of the model to observational world. Indeed, climate change is one of the main signal that we want to account for in the projected corrections. However, the change in the observations can be different, and therefore the resulting corrections can be also different from observations. Nevertheless, this methodology is justified because different simulations can have different variations, e.g., the four RCP scenarios provide four different simulations, giving four different corrections. This is also true for different climate models, which can show different changes. This information is therefore kept in the corrections."*

**3.3 page 7, line 18**

*Related to the previous item, " because we want to keep the evolution of the model" is a somewhat unscientific statement. Why do you want to assume that the time evolution is the same - what are the reasons that make this a suitable assumption here, especially for complex climate models?*

**3.3.1 Response**

The time evolution of the model represents a potential (biased) future of distributions of observations, constrained by the hypothesis over some natural and/or anthropogenic

forcing, and the physics of the model. We assume that the goal of bias correction is not to change these hypotheses and the physics, but to keep it (otherwise, why use a model?). According to our answer to the previous question, we cannot keep simultaneously the bias evolution and the time evolution. Thus, we kept the time evolution hypothesis.

**3.3.2 Modification**

See previous question/answer.

**3.4 page 8, line 16**

*" whereas it increases between ..." - It does, but only here for this example, not in general. Please mention this, to avoid confusion.*

**3.4.1 Response**

We agree.

**3.4.2 Modification**

The sentence *"...whereas it increases between $\mathbf{Y}^0$ and $\mathbf{Y}^1$."* has been corrected as follow: *"...whereas it increases between $\mathbf{Y}^0$ and $\mathbf{Y}^1$ in our example."*

**3.5  page 8, step 3**

*The proposed adaption seems somewhat unnatural to me. Looking at Figure 3, I would think that Figure 3a is a more appropriate reconstruction than Figure 3b, since it captures the important fact that the uncertainty about the values increases when transferring the assume dynamical evolution. This seems actually desirable! But of course it all depends on the goal here. Please comment!*

**3.5.1  Response**

Your suggestion (the increase of uncertainty) is very interesting, and offers a point of view that we had not considered. The reason we introduce this correction factor is that the transferred vectors do not form a reasonable transport plan because they can invert the quantiles. High temperatures can be transferred to low temperatures, even if the use of OTC after for the correction between $\mathbf{X}^1$ and $\mathbf{Y}^1$ could mask the problem.

**3.5.2  Modification**

The problem of quantile inversion has been added (p. 10, l. 2-3): *Furthermore, here the quantiles are inverted (low values are moved to high values)*

**3.6  Conclusion**

*Finally, it could be nice to discuss in the conclusions the relationship with copulas - which are functions that capture the dependence structure between two random variables $\mathbf{X}$, $\mathbf{Y}$, as does the transport plan here so there is an underlying " optimal transport copula" . Mentioning this connection could maybe make the work presented here interesting for a larger readership.*
**3.6.1 Response**

The link between copulas and optimal transport has been considered, but not treated.

**3.6.2 Modification**

A few words have been added in the perspective.

**4  Minor comments**

**4.1  page 3, line 13**

*Just a comment: the nomenclature is a bit strange (this seems to have historic roots in this field, so is not the authors' fault), the name " transfer function" does not seem a good choice as it means something quite different in dynamics. It would be more appropriate to simply call this a " map" .*

**4.1.1  Response**

"transfer function" means two different things in dynamical system theory and bias correction context. The term "transfer function" being associated to the map correcting the bias in bias correction, we prefer to keep this term.

**4.1.2 Modification**

To avoid confusion, we have been changed the sentence *"...is a transfer function $\mathcal{T}$ : $\mathbb{R}^d \rightarrow \mathbb{R}^d$..."* for *"...is a map $\mathcal{T} : \mathbb{R}^d \rightarrow \mathbb{R}^d$, called a* transfer function*,...".*

**4.2 page 3 , line 15**

*maybe add " deterministic" before " transfer function" ?*

**4.2.1 Response**

Agreed.

**4.2.2 Modification**

The change has been done.

**4.3 pager 12, line 19**

*" close" instead of " closed"*

**4.3.1 Response**

Indeed.

**4.3.2 Modification**

The change has been done.

**4.4 page 8, line 22-23**

*A matrix is not " definite" , so you probably mean " positive-definite" here in both cases?*

**4.4.1 Response**

Indeed.

**4.4.2 Modification**

The change has been done (p. 10, l. 11-13).

**4.5 page 14, line 25**

*" significant" instead of " significative"*

**4.5.1 Response**

Indeed.

**4.5.2 Modification**

The change has been done.

**4.6   page 14**

*The discussion of the results shown in Figure 5 is quite dense, it could benefit from a few more words?*

**4.6.1   Response**

The discussion has been detailed.

**Fig. 1.**

---

## Referee Report (RR1)

**Review of "Multivariate stochastic bias corrections with optimal transport" by Yoann Robin et al.**

Dr. Michael Muskulus, NTNU

This is a review of a revised manuscript that I have evaluated previously. I will keep my comments short thereof.

As stated earlier, the paper is interesting and should be published.

In the revisions, the authors have addressed all my comments. They have also included a simple univariate example to make the method more accessible to readers unfamiliar with the underlying theory, in response to the first reviewer. This improves the paper.

However, during the revision a few expressions were introduced that might be misunderstood. These should still be improved, for better readability, if possible.

Details:

- Figure 1, caption: The figure in b) shows probability levels $p_{x1}$, $p_{y1}$, $p_{x2}$, etc. on the vertical axes - these are difficult to read and more confusing than illuminating. I suggest to remove them. Also consider changing "indicate how the realizations of $x_1$ are distributed between each $y_j$" to "indicate the possibilities for how the probability of obtaining the value $x_1$ for X can be distributed among the possible values $y_j$ of Y". Maybe also picture even less than 100 arrows in panel d), as it is still quite crowded visually.
- page 4, line 15: Consider "Let $p_{xi}$ be the number ... in the interval representing $x_i$" instead of "Note $p_{xi}$ the number ... in the interval $x_i$", and similar for the next sentence.
- page 4, line 22: Consider "The black arrows represent the number of realizations $\gamma_{1j}$ that are transferred ..."
- page 5, line 21: Change "we note ... the set" to "let ... be the set"?
- page 5, line 22: Remove "=" sign after "is"
- Explanation in Section 2.1: This example uses realizations and the transport $\gamma$ is given in terms of *number* of realizations that are moved from a set of realizations of X to a same-size set of realizations of Y. In the next section the general theory is described, where the transport $\Gamma$ is now moving probabilities (e.g. such as to be independent of the number of realizations) between the laws of the random variables X and Y. This change from a set of realizations to random variables and their probability densities (actually probability measures, to be more correct) could confuse readers. It would be worth to use one sentence somewhere, e.g. at the end of Section 2.1, to explain this difference.

---

## Author Response (AR2)

**Response to review of "Multivariate stochastic bias corrections with optimal transport" by Dr. Michael Muskulus, NTNU**

Dr. Yoann Robin et al.

**Reviewer comment**

*This is a review of a revised manuscript that I have evaluated previously. I will keep my comments short thereof.*

*As stated earlier, the paper is interesting and should be published.*

*In the revisions, the authors have addressed all my comments. They have also included a simple univariate example to make the method more accessible to readers unfamiliar with the underlying theory, in response to the first reviewer. This improves the paper.*

*However, during the revision a few expressions were introduced that might be misunderstood. These should still be improved, for better readability, if possible.*

Dear Dr. Muskulus, thanks for your relevant comments. Your suggested modification have all been done to improve the paper.

**Details**

- *Figure 1, caption: The figure in b) shows probability levels $p_{x1}$, $p_{y1}$, $p_{x2}$, etc. on the vertical axes - these are difficult to read and more confusing than illuminating. I suggest to remove them. Also consider changing "indicate how the realizations of $x_1$ are distributed between each $y_j$" to "indicate the possibilities for how the probability of obtaining the value $x_1$ for X can be distributed among the possible values $y_j$ of Y". Maybe also picture even less than 100 arrows in panel d), as it is still quite crowded visually.*
  $\Rightarrow$ **The probability levels of Fig. 1b have been removed, the sentence have been modified, and we represent only 30 arrows.**
- *page 4, line 15: Consider "Let $p_{xi}$ be the number ... in the interval representing $x_i$" instead of "Note $p_{xi}$ the number ... in the interval $x_i$", and similar for the next sentence.*
  $\Rightarrow$ **The change has been done.**
- *page 4, line 22: Consider "The black arrows represent the number of realizations $\gamma_{1j}$ that are transferred ..."*
  $\Rightarrow$ **The change has been done.**
- *page 5, line 21: Change "we note ... the set" to "let ... be the set"?*
  $\Rightarrow$ **The change has been done.**
- *page 5, line 22: Remove "=" sign after "is"*
  $\Rightarrow$ **The change has been done.**
- *Explanation in Section 2.1: This example uses realizations and the transport $\gamma$ is given in terms of* number *of realizations that are moved from a set of realizations of X to a same-size set of realizations of Y. In the next section the general theory is described,*

*where the transport Γ is now moving probabilities (e.g. such as to be independent of the number of realizations) between the laws of the random variables X and Y. This change from a set of realizations to random variables and their probability densities (actually probability measures, to be more correct) could confuse readers. It would be worth to use one sentence somewhere, e.g. at the end of Section 2.1, to explain this difference.*

[revised manuscript text omitted]

---

## Author Response (AR3)

*Dear Authors,*

*Almost done, only a few technical aspects remain:*

*Code availability: Our policy is to make code required to reproduce analyses and results in a paper as openly available as possible. The standard nowadays is to publish the code on a repository of choice and name the repository in the paper. Can you do that, instead of making the code available upon request? If you have abolsutely no means to do that we won't enforce it, but it would surely support future use of your methods.*

*At the end of your text, there are two empty sections: Data availabaility / Code and data availabaility. Please remove them.*

*Yours sincerely, Uwe Ehret*

Dear Ehret,

The libraries Apyga and ARyga are now available on github, at:
https://github.com/yrobink/Ayga.git

The example of Section 3 is given in the example of Apyga. At the end of manuscript, the line "code and data availability" has been modified:

[revised manuscript text omitted]

Calibration          Projection

$$\mathbf{X}^0 \xrightarrow{\;\;\varphi\;\;} \mathbf{X}^1$$

$$\gamma \downarrow \qquad\qquad \downarrow \text{OTC}$$

$$\mathbf{Y}^0 \xrightarrow[\tilde{\varphi}]{\;\;\tilde{\varphi}\;\;} \mathbf{Y}^1 := \tilde{\varphi}(\mathbf{Y}^0)$$

**Figure 2.** Estimation of the unobserved random variable $\mathbf{Y}^1$. The random variables $\mathbf{X}^0$, $\mathbf{X}^1$ and $\mathbf{Y}^0$ are known. The plans $\gamma$ and $\varphi$ are the optimal joint laws in the sense of equations (2-4). $\tilde{\varphi}$ is the evolution of $\mathbf{Y}^0$ estimated from $\gamma$ and $\varphi$. OTC is used to correct $\mathbf{X}^1$ with respect to the estimation of $\mathbf{Y}^1$.

[Figure]

**Figure 3.** Bivariate histogram with bin size equal to 0.1. On each Panel we have: a Gaussian law centered in $(0,0)$ with covariance $4\mathbf{Id}_2$ ($\hat{\mathbb{P}}_{\mathbf{X}^0}$), a Gaussian law centered in $(10,0)$ with covariance $1/4\mathbf{Id}_2$ ($\hat{\mathbb{P}}_{\mathbf{X}^1}$) and a Gaussian law centered in $(0,10)$ with covariance $1/4\mathbf{Id}_2$ ($\hat{\mathbb{P}}_{\mathbf{X}^1}$). The red arrow is the local evolution between $\hat{\mathbb{P}}_{\mathbf{X}^0}$ and $\hat{\mathbb{P}}_{\mathbf{X}^1}$. (a) The probability distribution $\hat{\mathbb{P}}_{\mathbf{Y}^1}$ is the correction with OTC-$t$ and $\mathbf{D} = \mathbf{Id}_2$. The grey arrow is the estimation of the evolution of $\hat{\mathbb{P}}_{\mathbf{Y}^0}$. (b) The probability distribution $\hat{\mathbb{P}}_{\mathbf{Y}^1}$ is the correction with dOTC and $\mathbf{D}$ given by Eq. (6). The grey arrow is the estimation of the evolution of $\hat{\mathbb{P}}_{\mathbf{Y}^0}$.

[Figure]

**Figure 4.** Random variables generated by Lorenz (1984) model, OTC, dOTC, quantile mapping and CDF-$t$. (a) Biased random variable $\mathbf{X}^0$ (red) and references $\mathbf{Y}^0$ (blue) for time period 0. (b) Biased random variable $\mathbf{X}^0$ (red) and correction $\mathbf{Z}^0$ with OTC (green). (c) Biased random variable $\mathbf{X}^0$ (red) and correction $\mathbf{Q}^0$ with quantile mapping (green). (d) Biased random variable $\mathbf{X}^1$ (red) and references $\mathbf{Y}^1$ (blue) for time period 1. (e) Biased random variable $\mathbf{X}^1$ (red) and correction $\mathbf{Z}^1$ with dOTC (green). (f) Biased random variable $\mathbf{X}^1$ (red) and correction $\mathbf{Q}^1$ with CDF-$t$ (green).

[Figure]

**Figure 5.** (a) Map of the south east of the France. The 12 black squares are the locations where corrections are performed. (b-h) The $x$ axis of panels a-h is the evolution of the correction with dOTC. The $y$ axis of panels a-h is the evolution of WRF in red and the evolution of SAFRAN in blue. The red line is the linear regression between the evolution of correction and the evolution of WRF. The black cross marker are the scatter plot between the evolution of correction with CDF-$t$ and evolution of WRF. (b) Evolution of mean precipitation, i.e. difference between the projection period and the calibration period. (c) Evolution of variance of precipitation. (d) Evolution of spatial covariance of precipitation. (e) Evolution of covariance between precipitation and temperatures. (f) Evolution of mean temperatures. (g) Evolution of variance of temperatures. (h) Evolution of spatial covariance of temperatures.

**Table 1.** Representation of bias correction in context of climate change.

|  | Present | Future |
|---|---|---|
| Numerical model | $\mathbf{X}^0$ | $\mathbf{X}^1$ |
| Observations | $\mathbf{Y}^0$ | unknown ($\mathbf{Y}^1$) |

**Table 2.** $r$-value, $p$-value and standard error of linear regression between evolution of correction and evolution of WRF.

|  | $r$-value | $p$-value | standard error |
|---|---|---|---|
| Mean evolution Pr | 0.98 | $10^{-8}$ | 0.08 |
| Mean evolution Tas | 0.99 | $10^{-9}$ | 0.05 |
| Variance evolution Pr | 0.71 | $10^{-2}$ | 0.37 |
| Variance evolution Tas | 0.57 | $5 \times 10^{-2}$ | 0.25 |
| Covariance Pr/Tas evolution | 0.81 | $2 \times 10^{-3}$ | 0.24 |
| Spatial covariance Pr | 0.59 | $10^{-14}$ | 0.08 |
| Spatial covariance Tas | 0.76 | $10^{-29}$ | 0.05 |